# Peptide barcodes in dogs affected by mitral valve disease with and without pulmonary hypertension using MALDI-TOF MS and LC-MS/MS

Nattapon Riengvirodkij[1], Sittiruk Roytrakul[2], Janthima Jaresitthikunchai[2], Narumon Phaonakrop[2], Sawanya Charoenlappanich[2], Walasinee Sakcamduang[3]*

1 Prasu-Arthorn Animal Hospital, Faculty of Veterinary Science, Mahidol University, Nakhon Pathom, Thailand, 2 Functional Proteomics Technology Laboratory, National Center for Genetic Engineering and Biotechnology (BIOTEC), National Science and Technology Development Agency, Pathum Thani, Thailand, 3 Department of Clinical Sciences and Public Health, Faculty of Veterinary Science, Mahidol University, Nakhon Pathom, Thailand

* walasinee.sak@mahidol.ac.th

**Data Availability Statement:** All relevant data are within the manuscript.

## Abstract

Mitral valve disease (MVD) is an important and most frequently acquired heart disease found in dogs. MVD is classified into different stages according to its severity. There is a challenge in differentiation between asymptomatic and symptomatic stages of the MVD. Moreover, pulmonary hypertension (PH) is a common complication in dogs affected by MVD. In clinical practice, there are also some limitations to identify PH. Matrix-assisted laser desorption/ionization time-of-flight mass spectrometry (MALDI-TOF MS) is a technique that can characterize specific patterns of peptide mass called peptide barcodes from various samples. Besides, in combination with liquid chromatography-tandem mass spectrometry (LC-MS/MS), potential peptide sequences associated with specific conditions could be identified. The present study aimed to use MALDI-TOF coupled with LC-MS/MS to characterize specific peptide barcodes and potential peptide candidates in serum samples from healthy dogs, dogs with MVD stage B (MVD B, asymptomatic stage), MVD stage C (MVD C, symptomatic stage), MVD stage B with PH (MVD B PH), and MVD stage C with PH (MVD C PH). Discrete clusters of the 5 sample groups were identified by 3D plot analysis. Peptide barcodes also revealed differences in peptide patterns among the 5 groups. Six amino acid sequences of peptide candidates at 1,225.60, 1,363.85, 1,688.71, 1789.52, 2020.21, and 2156.42 Da were identified as part of the proteins CLCN1, CLUL1, EDNRA, PTEN, SLC39A7, and CLN6, respectively. The network interactions between these discovered proteins and common cardiovascular drugs were also investigated. These results demonstrate that MALDI-TOF MS has promise as an optional technique for diagnosing dogs affected by asymptomatic and symptomatic stages of MVD with and without PH. Further studies are required to identify peptide barcodes in dogs with other diseases to create peptide barcode databases in veterinary medicine before using this method as a novel diagnostic tool in the future.

**Funding:** This study was supported by a research assistance (RA) scholarship from the Faculty of Graduate Studies, Mahidol University Academic Year 2019. The funding was mainly used for clinical procedures, including echocardiography, radiography, electrocardiography, hematology, and biochemical profiles. The funder had no involvement in the design, sample collection, and analysis of the study.

**Competing interests:** The authors have declared that no competing interests exist.

## Introduction

Mitral valve disease (MVD) is the most common and important acquired heart disease in dogs. It is highly prevalent in small-breed, particularly in older dogs, with a prevalence of up to 90% in small-breed dogs older than 8 years old [1]. The disease is characterized by grossly thickening and deformation of the mitral valve with prolapsed leaflets in many cases. The etiology of MVD is currently unclear; however, it is believed that abnormalities of collagen and extracellular matrix within the valves are involved in the pathogenesis of the disease [2]. Some studies have been suggested that genetic predisposition is associated with the disease in the Cavalier King Charles Spaniel (CKCS) [3, 4]. MVD is classified in different stages according to its severity, and the differentiation between asymptomatic and symptomatic stages of the MVD is very challenging in clinical practice. The diagnosis of symptomatic MVD is based on clinical signs of left-sided congestive heart failure (CHF) from physical examination, such as respiratory distress, tachypnea, and cough [5]. These signs are nonspecific and could also be found in patients with concomitant respiratory diseases. As observed by thoracic radiography, the presence of pulmonary edema in the perihilar region is generally used for detecting left-sided CHF. However, this procedure should be avoided when patients encounter severe respiratory distress.

Pulmonary hypertension (PH) is a pathologic condition defined as increased blood pressure in the pulmonary vascular system [6, 7]. PH can occur as primary (idiopathic) or secondary PH, which could have various underlying causes [8, 9], but the most common underlying cause of PH is MVD [6, 10]. The gold standard for the definitive diagnosis of PH is cardiac catheterization of the right heart to measure the pulmonary arterial pressure (PAP) directly in the pulmonary vasculature [11], which is not practical in general veterinary practice. Alternatively, echocardiography is promising in the diagnosis of PH [12, 13]. However, this technique has some limitations, including; the need for flow velocity of tricuspid or pulmonic regurgitation, which can be absent in some patients, poor alignment of the interrogation beam to the blood flow direction, poor cooperation of patients during the procedure, and skill and experience of sonographers [10].

Determination of the N-terminal pro B-type-natriuretic peptide (NT-proBNP) level has been added as an adjunctive test for diagnosing both symptomatic MVD and PH. Unfortunately, the NT-proBNP concentration alone is insufficient to be used for identifying PH [5, 14, 15], and differentiating symptomatic from subclinical MVD [16, 17]. In recent years, Matrix-assisted laser desorption/ionization time-of-flight mass spectrometry (MALDI-TOF MS) has been introduced as a novel instrument used in clinical diagnosis in both human and veterinary medicine due to its rapidity of interpretation, high efficacy, and cost-effectiveness [18]. MALDI-TOF MS is a potent method with great capability to identify peptide mass patterns called peptide barcodes in many types of samples (e.g., serum, saliva, sweat, and tissue). The obtained peptide barcode is typically unique to each condition or disease. Therefore, these peptide barcodes can be used to recognize diseases and serve as alternative diagnostic tools in clinical practice. At present, MALDI-TOF MS has been applied in human medicine as a diagnostic test, especially in the field of neoplastic diseases, for early diagnosis, monitoring, and mechanistic determination. It is used to detect numerous types of cancer in the early stage of disease, including gastrointestinal cancer [19], lung cancer [20], renal cell carcinoma [21], bladder cancer [22], prostate cancer [23, 24], breast cancer [25–27], and leukemia [28]. In veterinary medicine, MALDI-TOF MS has been evaluated to characterize unique peptide barcodes in dogs with different oral tumors using oral tumor tissues [29], and saliva samples [30]. Another technique that is commonly used in peptidomic studies is liquid chromatography tandem mass spectrometry (LC-MS/MS). This method has the ability to identify peptide sequences from

various samples. To our knowledge, the study of peptide barcodes and the identification of potential peptide sequences using MALDI-TOF coupled with LC-MS/MS in dogs with MVD with or without PH has never been evaluated. The present study aimed to demonstrate peptide barcode and peptide candidates associated with each condition that could potentially be developed as novel diagnostic techniques obtained from serum samples in healthy dogs (normal control group), dogs with MVD stage B (MVD B group, asymptomatic stage), MVD stage C (MVD C group, symptomatic stage), MVD stage B with PH (MVD B PH group) and MVD stage C with PH (MVD C PH group).

## Materials and methods

### Animals

Blood samples were collected from dogs submitted as clinical cases at Prasu-Arhtorn Animal Hospital, Faculty of Veterinary Science, Mahidol University, Thailand. Informed consent was obtained from all owners before their dogs were enrolled in the study. The protocol used in this study was approved by Mahidol University-Institute Animal Care and Use Committee of the Faculty of Veterinary Science., number MUVS-2017-12-57. The study was prospectively conducted from March 2018 to August 2019. Dogs were required to be at least 5 years old with no limit on sex, breed, or weight for inclusion in the study. All dogs underwent history taking and a complete physical examination. The cardiorespiratory system was evaluated using thoracic auscultation to check for the presence/absence of a heart murmur and the type, type, location, and intensity (grade I-VI) of the heart murmur.

Thoracic radiographs were obtained in right lateral and ventrodorsal (VD) or dorsoventral (DV) recumbency for all recruited dogs. Evidence of cardiomegaly, left- and/or right-sided heart enlargement, pulmonary venous congestion, main pulmonary arterial dilatation, and pulmonary edema was evaluated. Electrocardiography (ECG) was performed using a standard six-lead recording system in right lateral recumbency. Six-lead ECG data were recorded for approximately 10 seconds in each dog. *F*urther one-minute recording of the cardiac rhythm by lead II was performed if arrhythmia was detected.

The echocardiographic examination was performed by one well-trained investigator using a GE Vivid E9 ultrasound machine with a multi-frequency sector transducer (4.5–12 MHz probe) with continuous ECG recording during the process. The enrolled dogs were not sedated during the procedure, and all measurements were repeated for at least three consecutive cardiac cycles. The right parasternal (RPS) long-axis view on 2-dimensional (2D) echocardiography was used to assess valve structure and function, including valve degeneration, valvular prolapse, and chordae tendineae rupture. Colour flow Doppler was used to identify the presence of valvular regurgitation and consider the degree of mitral regurgitation (MR) semi-quantitatively. The left atrial-to-aorta (LA/Ao) ratio was measured during diastole in the RPS short-axis 2D view to identify the presence of left atrial dilatation [31]. The main pulmonary artery (PA) size was also assessed in the 2D RPS short axis. Short-axis M-mode echocardiographic examination of the left ventricle was used to demonstrate the percentage of fractional shortening (FS%) and determine the left ventricular internal diameter in the diastolic phase (LVIDd), and the left ventricular internal diameter in the systolic phase (LVIDs). To normalize these values based on body weight, the LVIDd (cm) was divided by (body weight (kg))$^{0.294}$ and the LVIDs (cm) was divided by (body weight (kg))$^{0.315}$ to obtain the normalized left ventricular internal diameter in the diastolic phase (NLVIDd) and the normalized left ventricular internal diameter in the systolic phase (NLVIDs), respectively [32].

The left parasternal (LPS) apical 4-chamber view, LPS long-axis view of the right auricle, and LPS cranial transverse view of the tricuspid valve that provided the optimal alignment of

the continuous wave beam and tricuspid regurgitation (TR) flow were used to measure the maximal flow velocity of the TR jet. Peak TR flow measurement was applied using the modified Bernoulli equation (Pressure gradient (PG) = 4 x velocity$^2$) to calculate the systolic PG across the tricuspid valve, representing the systolic PAP. A peak TR jet velocity higher than 2.8 m/s and a PAP equal to or greater than 31 mmHg indicate PH. Pulmonary stenosis was ruled out before the diagnosis of PH was made decisively by measuring the peak pulmonary artery flow velocity, which should be lower than 1.5 m/s without turbulent flow across the pulmonary valve.

Dogs were excluded from the study if they were found to have other significant concurrent systemic diseases, such as renal disease, hepatic disease, hormonal disease, or gastrointestinal disease, which may interfere with protein expression. Dogs with any other congenital or acquired heart diseases other than MVD were also excluded from the study. However, dogs that received standard therapy to stabilize congestive heart failure, such as diuretics, angiotensin-converting enzyme inhibitors (ACEIs), inotropic agents, and vasodilators, were all permitted to participate in this study.

Fifty-nine dogs were enrolled in the study. They were classified into 5 groups according to American College of Veterinary Internal Medicine (ACVIM) classification from the guidelines for the diagnosis and treatment of mitral valve disease in dogs [33] and further subdivided according to the presence/absence of PH.

**The MVD B group** included 16 dogs with MVD stage B2, i.e., dogs affected by MVD with no clinical signs of congestive heart failure (asymptomatic patients). No radiographic evidence of congestive heart failure and no evidence of pulmonary edema and/or venous congestion was detected, but there was sufficient hemodynamic change to cause echocardiographic evidence of left atrial dilatation with an LA/Ao ratio greater than 1.6 [31], and left ventricular enlargement with a LVIDDN greater than 1.7 [32].

**The MVD B PH group** included 5 dogs affected by MVD stage B2 with PH, i.e., dogs affected by MVD with secondary PH but with no clinical signs of congestive heart failure (asymptomatic patients). The criteria for classifying MVD stage B were similar to those mentioned above in group MVD B. Dogs with PH were classified by Doppler echocardiography assessing a peak TR velocity greater than or equal to 2.8 m/s and/or pulmonary regurgitant velocity greater than or equal to 2.2 m/s [13].

**The MVD C group** included 11 dogs with MVD stage C, i.e., dogs affected by MVD with echocardiographic evidence of MVD and clinical signs of congestive heart failure (symptomatic patients), identified by clinical examination and radiographic evidence of pulmonary edema and/or pulmonary venous congestion.

**The MVD C PH group** included 16 dogs affected by MVD stage C with PH, i.e., dogs affected by MVD with secondary PH with clinical signs of congestive heart failure (asymptomatic patients). The criteria for classifying MVD stage C and PH were similar to those mentioned above.

**The normal control group** comprised 11 normal healthy control dogs with no heart disease or PH, referred to as the NC group. These had no history or clinical signs of cardiorespiratory diseases, such as coughing, dyspnea, exercise intolerance, cyanosis, or syncope. No heart murmurs or crackle lung sounds were detected by auscultation. Each animal was inspected by thoracic radiography, ECG, and echocardiography to ensure that the animal had a normal heart condition. The age of the dogs enrolled in this group was matched with that of the dogs in the other groups.

Baseline characteristics of all dogs were compared among the 5 groups using SPSS software version 24. A normal distribution could not be assumed due to the small sample size; therefore,

all numeric variables were analyzed with nonparametric tests. The Kruskal-Wallis test was used to analyze continuous variables, and the chi-square test was used for categorical variables.

## Sample preparation

Five milliliters of blood were taken from the jugular vein or saphenous vein for routine hematology and serum biochemistry. The first portion, consisting of 0.5 ml, was transferred into an ethylenediaminetetraacetate (EDTA) tube for a complete blood count (CBC). The second portion, consisting of 1.5 ml, was transferred into a heparinized tube for biochemical analysis, including the determination of alanine aminotransferase (ALT), alkaline phosphatase (ALP), creatinine, blood urea nitrogen (BUN), total protein (TP), and albumin. The remaining 3 ml of blood was stored in a serum tube for the peptidomic process. Serum samples were separated by centrifugation at 4˚C and 3000 g for 10 minutes within 30 minutes after collection. Subsequently, each sample was aliquoted and kept at -80˚C for subsequent analyses. The total protein concentration in each serum sample was evaluated by Lowry's assay at 690 nm using bovine serum albumin (BSA) as a standard [34].

## Analysis of serum peptides by MALDI-TOF MS

The serum samples in each group were prepared at a 1 mg/ml concentration in 0.1% trifluoroacetic acid (TFA). The whole serum was pooled for 10 μL per sample at a concentration of 1 mg/ml. The pooled samples were then mixed with MALDI matrix solution consisting of 10 mg/ml α-cyano-4-hydroxycinnamic acid (CHCA) in 100% acetonitrile (ACN) containing 5% TFA at a ratio of 1:3 (sample: matrix), and 2 μL of the mixed samples was directly spotted on the MALDI steel target plate (MTP 384 ground steel, Bruker Daltonics, Billerica, Massachusetts, USA) with 36 replicates. After air drying, mass spectra were developed by an Ultraflex III TOF/TOF system (Bruker Daltonics) in a linear positive mode with a mass range of 1,000–20,000 Da. External calibration was performed using the Proteo-Mass Peptide & Protein MALDI MS Calibration Kit (Sigma Aldrich, St. Louis, Missouri, USA), including human angiotensin II (1,046.5423 m/z), synthetic peptide P14R (1,533.8582 m/z), human ACTH fragment 18–39 (2,465.1989 m/z), bovine insulin oxidized B chain (3,494.6513 m/z), bovine insulin (5,735 m/z), equine cytochrome c (12,362 m/z), and equine apomyoglobin (16,952 m/z). The mass spectra of peptide barcodes were analyzed by Flex Analysis version 3.3 software (Bruker Daltonics). Dendrograms and three-dimensional principal component analysis (3D PCA) scatterplots were analyzed by ClinPro Tools version 3.0 software (Bruker Daltonics) [35]. To analyze the differences in the peptide mass spectra among the groups, the Wilcoxon test was used. A P-value < 0.05 was considered statistically significant.

## Peptide identification by LC-MS/MS

Peaks of peptide mass spectra with high signal intensities were further selected to be evaluated by LC-MS/MS to identify specific peptide sequences. Serum samples were purified by C18 Zip-Tip (Merck Millipore, Darmstadt, Germany) and diluted with 2% ACN. Next, serum samples were analyzed by LC-MS/MS. An ultimate 3000 LC system (Thermo Scientific Dionex, Waltham, MA, US) on a nanocolumn (PepSwift monolithic column, 100 μm in diameter x 50 mm in length) connected to an electrospray ionization system in positive ion mode and a Hybrid quadrupole Q-TOF impact II™ system (Bruker Daltonics GmbH, Germany) was used. MaxQuant (version 1.6.6.0) was used for peptide identification against the UniProt data for *Canis lupus familiaris* database [36]. From the UniProt data, the identified peptides were then be recognized as part of expected proteins. The relationship of these identified proteins and common cardiovascular drugs was demonstrated using Stitch version 5.0 [37].

## Results

Fifty-nine dogs were enrolled in the study. They were classified into 5 groups according to the ACVIM classification [34], including 16 dogs with MVD stage B2 (MVD B), 5 dogs with stage B2 and PH (MVD B PH), 11 dogs with MVD stage C (MVD C), 16 dogs with MVD stage C and PH (MVD C PH), and 11 normal control dogs. The MVD B group (n = 16) consisted of 2 Shih-Tzus, 5 Poodles, 3 mixed-breed dogs, 3 Pomeranians, 2 Chihuahuas, and 1 Dachshund. The MVD C group (n = 11) included 3 Chihuahuas, 2 Poodles, 3 mixed-breed dogs, 2 Shih-Tzus, and 1 Pomeranian. The MVD B PH group (n = 5) comprised 3 Chihuahuas, 1 Poodle, and 1 Shih-Tzu. The MVD C PH group (n = 16) consisted of 9 Poodles, 2 Chihuahuas, 2 Pomeranians, and 1 each of the following breeds: Shih-Tzu, Golden Retriever, and mixed-breed dog. The normal control group (n = 11) included 2 Poodles, 8 Chihuahuas, and 1 Pomeranian. The baseline characteristics of all 59 dogs are reported in Table 1. The median body weight, proportion of females versus males, and FS% were not significantly different among the groups. In contrast, the median age, LA:Ao ratio, NLVIDd, and NLVIDs were shown to be significantly different among the groups. The normal dogs were significantly younger than the dogs in the other groups. Compared with normal dogs, dogs with MVD stage B, MVD stage C, and MVD stage C with PH had a significantly increased LA:Ao ratio, NLVIDd, and NLVIDs. Dogs with MVD stage C with or without PH had an increased LA:Ao ratio compared to dogs with MVD stage B with or without PH. Dogs with MVD stage C showed a significantly higher NLVIDd and NLVIDS than dogs with MVD stage B with and without PH.

### Peptide barcodes by MALDI-TOF

The 3D-PCA scatterplot revealed distinct clusters among the MVD B, MVD C, MVD B PH, MVD C PH, and normal control groups. All 36 replicates from each pooled serum sample group exhibited a clearly distinguished cluster from the others, indicating a distinctive peptide profile in each individual group and demonstrating the uniformity and homogeneity of data within the groups (Fig 1). Thirty-six dots with different colors represent the replicated pooled serum samples within each group. Different peptide barcodes in the MVD B, MVD C, MVD B PH, MVD C PH, and normal control groups were identified with a detection range of 1,000–20,000 Da (Fig 2). The MALDI-TOF MS results indicated accurate outcomes with 95% confidence intervals. The cross-validation result, calculated by ANOVA, in the MVD B, MVD B PH, MVD C, MVD C PH, and normal groups was 93.33%, 100%, 100%, 100%, and 100%,

**Table 1. Baseline characteristics of the 59 dogs in all groups.**

| Variables | MVD B (n = 16) | MVD B PH (n = 5) | MVD C (n = 11) | MVD C PH (n = 16) | Normal (n = 11) | P value |
|---|---|---|---|---|---|---|
| Age (years) | 11.8[a] (9.1–14.3) | 12.0[a] (8.3–13.5) | 10.3[a] (10–12.5) | 11.8[a] (10.8–12.9) | 5.0[b] (5–7.79) | **0.003** |
| Body weight (kg) | 5.4 (4.05–8.35) | 5.3 (4–5.8) | 4.7 (4.25–6.8) | 5.45 (4.3–7.78) | 3.7 (3.15–5.15) | 0.233 |
| Female (percent) | 6/16 (37.5%) | 2/5 (40%) | 5/11 (45.5%) | 6/16 (37.5%) | 7/11 (63.6) | 0.679 |
| LA:Ao ratio | 1.69[a] (1.5–1.97) | 1.70[ac] (1.22–1.88) | 2.06[b] (1.83–2.21) | 2.01[b] (1.95–2.69) | 1.27[c] (1.11–1.37) | **<0.001** |
| NLVIDd (cm) | 1.71[a] (1.49–1.88) | 1.49[ac] (1.33–1.58) | 1.96[b] (1.79–2.11) | 1.91[ab] (1.56–2.31) | 1.36[c] (1.28–1.41) | **<0.001** |
| NLVIDs (cm) | 0.81[a] (0.75–0.92) | 0.74[a] (0.63–0.83) | 0.97[b] (0.89–1.09) | 0.94[ab] (0.79–1.12) | 0.74[ac] (0.66–0.82) | **0.024** |
| FS (%) | 48.38 (44.49–56.74 | 48.51 (48.32–42.38) | 45.35 (41.77–54.53) | 50.38 (46.32–54.45) | 42.11 (38.86–48.94) | 0.289 |

Data are reported as the median (Q1-Q3), and the female sex is reported as the proportion (percent).

Bold p values indicate significance (P-value < 0.05).

Within the same row, values with the same letter in superscript do not differ significantly (P value >0.05).

LA:Ao left atrium-to-aorta ratio, NLVIDd normalized end-diastolic left ventricular internal diameter, NLVIDs normalized end-systolic left ventricular internal diameter, FS% fractional shortening percentage.

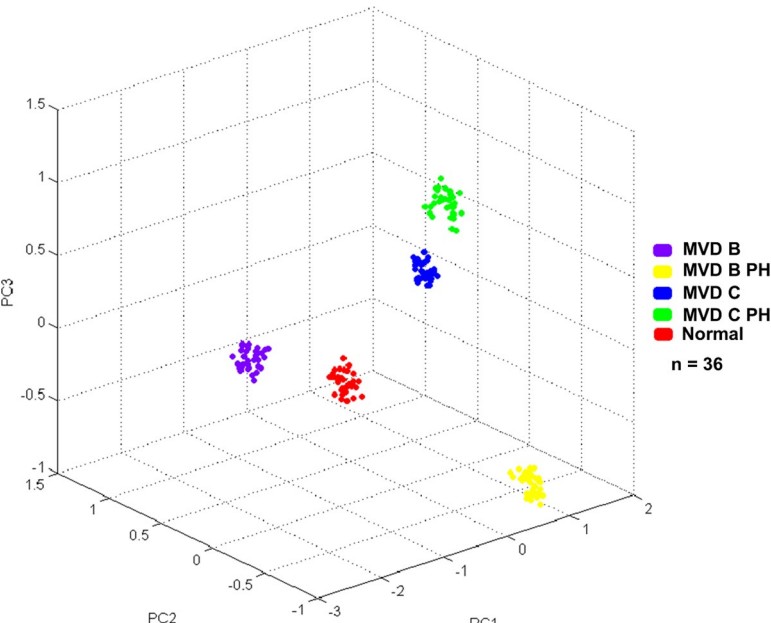

**Fig 1. Three-dimensional principal component analysis (3D-PCA) scatterplot of dogs in the MVD stage B2 (MVD B), MVD stage B2 with PH (MVD B PH), MVD stage C (MVD C), MVD stage C with PH (MVD C PH), and normal control groups.**

respectively. The recognition capability calculated by QC/Different Average, SNN, AD, TTA, W/KW, and Genetic Algorithm in the MVD B, MVD B PH, MVD C, MVD C PH, and normal groups was 100% for each, indicating that the results were of high reliability.

Different mass spectral peaks of peptide barcodes among all 5 groups were selected and demonstrated by ClinPro Tools software, including peptide A with a mass spectral peak at 1,225.60 Da, peptide B with a mass spectral peak at 1,363.85 Da, peptide C with a mass spectral peak at 1,688.71 Da, peptide D with a mass spectral peak at 1,789.52 Da, peptide E with a mass spectral peak at 2,020.21 Da, peptide F with a mass spectral peak at 2,156.42 Da (Fig 3).

## Peptide identification by MALDI-TOF and LC-MS/MS

Mass spectral peaks with high signal intensities in all sample groups were further analyzed by LC-MS/MS. Peptide A, with a mass spectral peak at 1,225.60 Da, was identified as part of chloride channel protein 1 (CLCN1). Peptide B, with a mass spectral peak at 1,363.85 Da, was identified as part of clusterin-like protein 1 precursor (CLUL1). Peptide C, with a mass spectral peak at 1,688.71 Da, was identified as part of endothelin-1 receptor precursor (EDNRA). Peptide D, with a mass spectral peak at 1,789.52 Da, was identified as part of phosphatidylinositol-3,4,5-trisphosphate 3-phosphatase and dual-specificity protein phosphatase (PTEN). Peptide E, with a mass spectral peak at 2,020.21 Da, was identified as part of zinc transporter SLC39A7 (SLC39A). Peptide F, with a mass spectral peak at 2,156.42 Da, was identified as part of ceroid-lipofuscinosis neuronal protein 6 homolog (CLN6) (Table 2). Networks of protein-protein and protein-cardiovascular drug interactions were evaluated by the Stitch program, version 5.0. The strength of these pathway interactions at the functional level was assessed by edge confidence scores. Interactions with high edge confidence scores (>0.700) are represented as thick lines, indicating strong relationships for the protein-protein and/or protein-cardiovascular drug interactions. Four out of 6 proteins, including CLCN1, EDNRA, PTEN and SLC39A7,

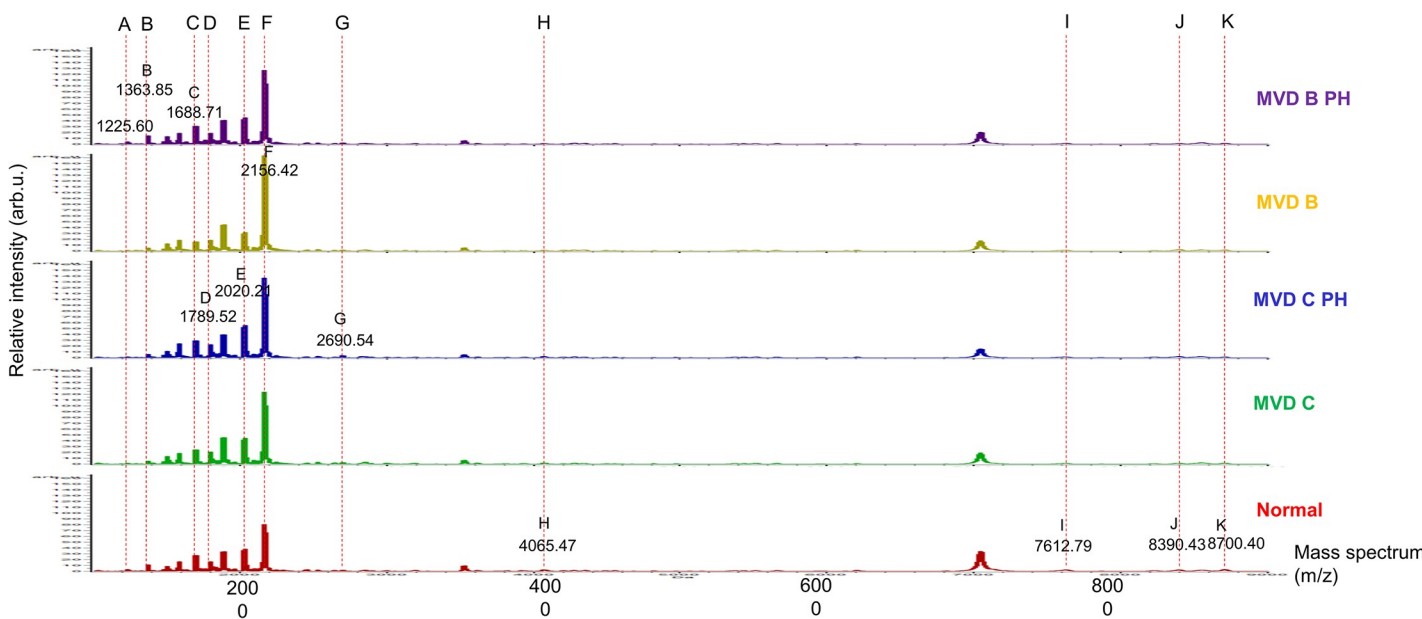

**Fig 2. Peptide barcodes of dogs in the MVD stage B2 with PH (MVD B PH), MVD stage B2 (MVD B), MVD stage C with PH (MVD C PH), MVD stage C (MVD C), and normal control groups in the detection range 1,000–20,000 Da.**

revealed strong relationships with cardiovascular drugs that are commonly used clinically. These include furosemide, ramipril, benazepril, imidapril, pimobendan, spironolactone, and sildenafil. The remaining 2 proteins, CLUL1 and CLN6, showed no association or interaction with any cardiovascular drug (Fig 4).

## Discussion

To our knowledge, this is the first study to analyze peptide barcodes using the MALDI-TOF MS method in dogs affected by different stages of MVD with and without PH. The differentiation between asymptomatic and symptomatic MVD in dogs is very challenging due to the nonspecificity of the clinical signs. In addition, echocardiography, the conventional method for identifying both MVD and PH in dogs, may have some clinical limitations in practical use [10]. Although echocardiography is currently a method that most suitable for diagnosis of MVD and PH in clinical practice, peptide barcodes from MALDI-TOF MS may provide additional information in identifying dogs with MVD and PH and could serve as helpful alternative or optional diagnostic technique of MVD and PH in the future. MALDI-TOF MS is a promising method for detecting MVD with and without PH in dogs by using serum samples, which are easy to collect; furthermore, the results do not require expert skill for interpretation. It is a fast, accurate, and reliable method with high sensitivity and high reproducibility, making it appropriate for the clinical diagnosis of MVD and PH in dogs. The analysis of peptides with MALDI-TOF MS also allows multiple groups of peptides to be identified at once. In contrast, conventional methods, such as ELISA, can only analyze one type of peptide at a time. For this reason, peptide barcodes derived from MALDI-TOF MS have more potential for diagnosing diseases than conventional ELISA techniques. Peptide barcodes are determined by an analytical technique for demonstrating different peaks of peptides that are usually unique to each condition (e.g., disease) or organism (e.g., bacteria, fungi or plants). The present study demonstrated differences in the serum peptide barcodes of dogs in the MVD B, MVD C, MVD B PH, MVD C PH, and normal control groups using MALDI-TOF MS. Peptides derived from

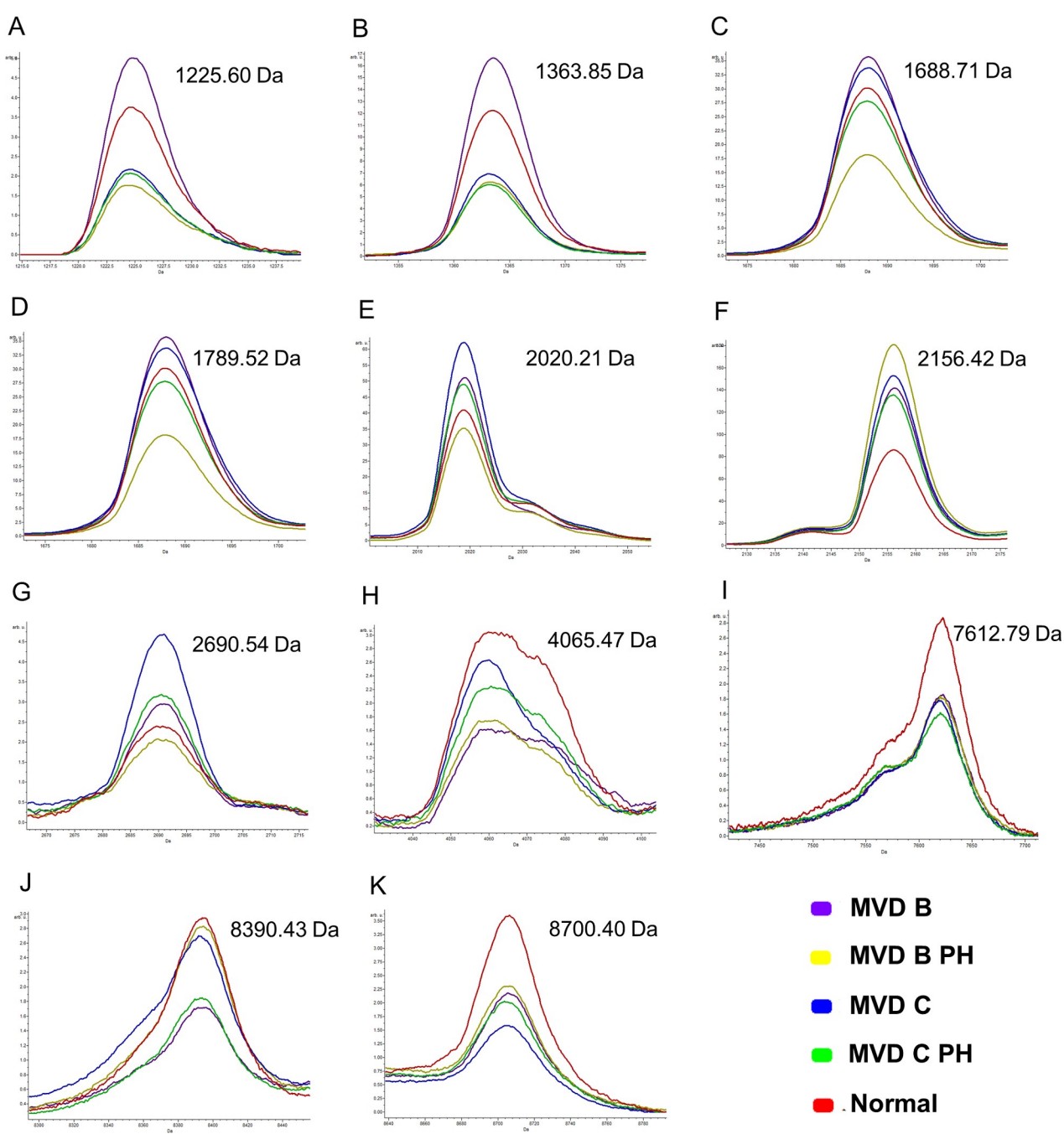

**Fig 3. Magnified view of different mass spectral peaks of peptide barcodes among all 5 groups from Fig 2.**

proteins were altered in their abandon, representing changes in the protein levels of peptide barcode components among the groups. Additionally, the scatterplot analysis by MALDI-TOF MS exhibited discrete clusters of peptide expression in each sample group. According to these results, MALDI-TOF has the potential for use as a rapid and efficient method for detecting PH and differentiating between symptomatic and asymptomatic MVD.

Peptide barcodes obtained from MALDI-TOF MS have been reported as an alternative tool for detecting human breast cancers. Kang et al. [38] investigated the protein MALDI profile of

**Table 2. Amino acid sequence of peptide candidates from Fig 4 analysed by LC-MS/MS.**

| Peptide | Mass (Da) | Amino acid sequence | Expected protein | |
|---|---|---|---|---|
| | | | UniProt accession No. | Protein name |
| A | 1225.689 | GLRANTRPTQI | Q9MZT1 | Chloride channel protein 1 (ClC-1) (CLCN1) |
| B | 1363.768 | TNLMKTLKKCK | Q95KN1 | Clusterin-like protein 1 (retinal-specific clusterin-like protein) (CLUL1) |
| C | 1688.939 | GDLIYVVIDLPINVF | Q5KSU9 | Endothelin-1 receptor (endothelin receptor type A) (ET-AR) (EDNRA) |
| D | 1789.007 | IKVEFFHKQNKMLK | P60483 | Phosphatidylinositol 3,4,5-trisphosphate 3-phosphatase and dual-specificity protein phosphatase (PTEN) |
| E | 2020.193 | LLREASPLQSLLEVLGLLG | Q5TJF6 | Zinc transporter SLC39A7 (SLC39A7) |
| F | 2156.215 | LWNDPVLRKKYPGVIYVP | Q5JZQ8 | Ceroid-lipofuscinosis neuronal protein 6 homologue (CLN6) |

34 pairs of resected breast cancer and adjacent normal tissue samples. The results showed that the peptide profile could noticeably discriminate breast cancer from normal tissue samples. In another study, MALDI-TOF MS was used to create a peptide profile reference for the classification of various cancer cell lines, such as human melanoma, human breast carcinoma and human liver carcinoma cell lines. The results revealed different peptide barcodes in each cancer cell line that could be used to identify cancer types in patients clinically [39]. In veterinary medicine, peptide barcodes have also been demonstrated to be used for classifying different types of canine oral tumors using tumor tissue samples [29] and salivary samples [30]. A study using peptide barcodes for the diagnosis of canine heart disease has never been performed; thus, this study is the first to define peptide barcodes specific to different stages of MVD with and without PH in dogs.

NT-proBNP has been defined as a cardiac biomarker for diagnosing, therapeutic monitoring and prognosis of MVD in dogs [16, 17, 40–43]. NT-proBNP is an amino-terminal fragment of prohormone, proBNP, consisting of 76 amino acids [44]. The exact molecular weight of NT-proBNP has never been reported. However, using a tool on the ExPASy server (https://web. expasy.org/compute_pi/pi_tool-doc.html), the molecular weight of NT-proBNP can be calculated as approximately 8,087 Da. However, no mass spectral peak at this molecular weight was detected in our samples; therefore, this could indicate that peptide sequences within all mass spectral peaks at different molecular weights were not NT-proBNP. Another cardiac biomarker that has been clinically used for identifying heart disease is cardiac troponin I (cTnI). In veterinary medicine, an elevated cTnI concentration suggests the presence of myocardial injury [45]. However, it does not classify the cause of injury. cTnI has also been reported to be significantly increased in dogs with pre-capillary PH compared to normal dogs and dogs with MVD [15]. cTnI has a molecular weight of 24,000 Da [46], which is larger than all mass spectral peaks obtained from this study. No peptide with a similar sequence to cTnI and NT-proBNP was detected by LC-MS, implying that peptides detected by MALDI-TOF might be used as new cardiac biomarkers.

NT-proBNP is produced inside ventricular myocytes by the cleavage of pro-BNP, a 108-amino acid prohormone, via 2 proteolytic enzymes, furin and corin. The cleavage of proBNP is believed to occur during the peptide secretion process [47]. cTnI is an intracellular cardiac protein that involves the troponin complex to prevent an interaction between actin and myosin during cardiac muscle relaxation. This protein is released into the circulation when cardiomyocyte injury occurs. Novel peptides that we could identify from each mass spectral peak in the present study may be synthesized and secreted into the circulation by a process similar to that of these 2 previous peptides.

Amino acid sequences of peptide candidates from discriminatory mass spectral peaks were analyzed by LC-MS/MS. Six amino acid sequences of peptide candidates were identified (peptides A-F), as shown in Table 2. These obtained peptide candidates with mass spectral peaks at

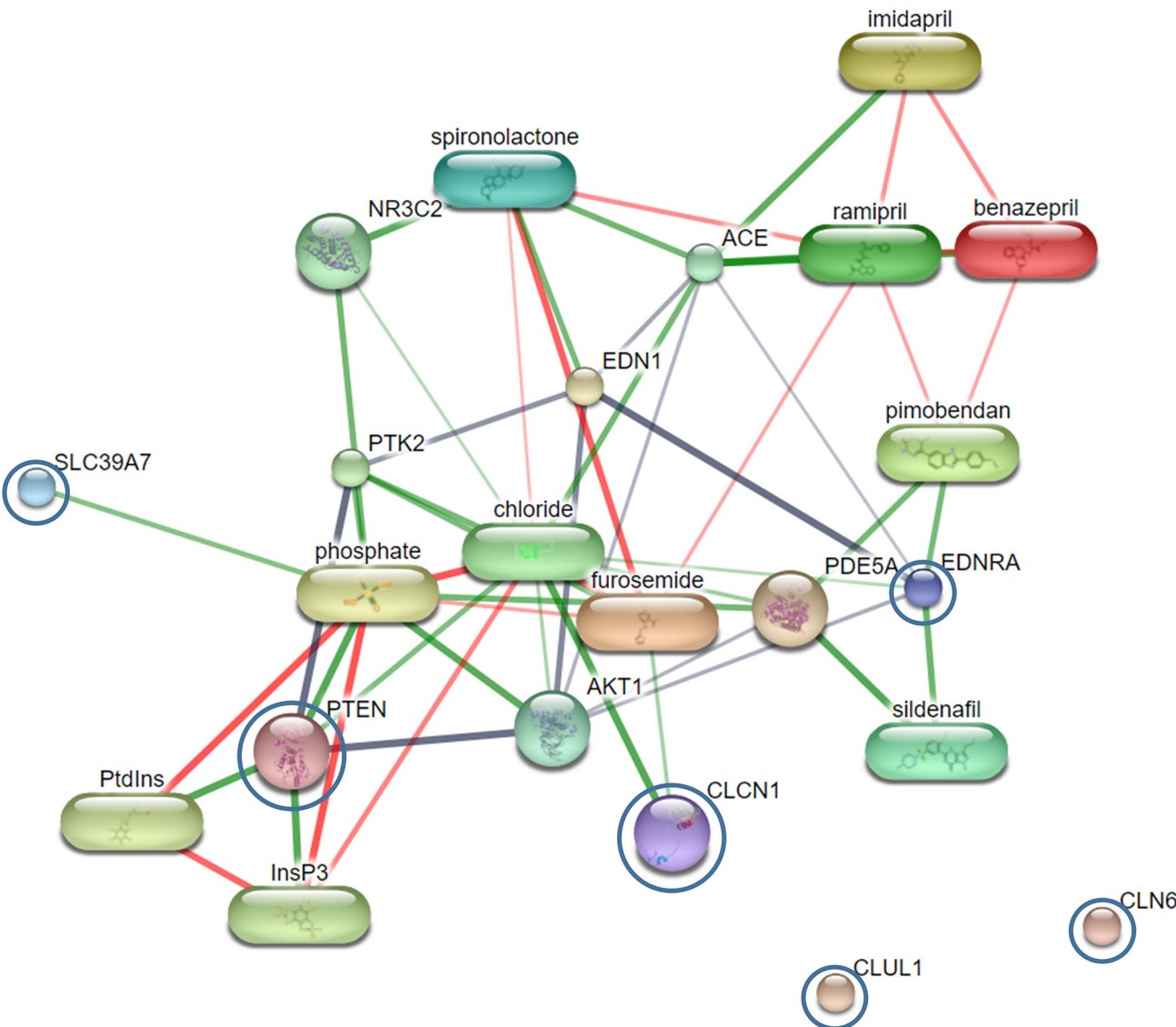

**Fig 4. Relationships of CLCN1, EDNRA, PTEN, SLC39A7, CLUL1 and CLN6 proteins (blue circles) in the network of protein-cardiovascular drug interactions.** Abbreviations: ACE, angiotensin-converting enzyme; AKT1, v-akt murine thymoma viral oncogene homologue 1; CLCN1, chloride channel protein 1; CLN6, ceroid-lipofuscinosis neuronal protein 6 homologue; CLUL1, clusterin-like protein 1 precursor; EDN-1, endothelin-1; EDNRA, endothelin-1 receptor precursor; InsP3, intracellular messenger formed by the action of phospholipase C on phosphatidylinositol 4,5-bisphosphate; PDE5A, phosphodiesterase 5; PtdIns, phosphatidic acid combined with inositol; PTEN, phosphatidylinositol-3,4,5-trisphosphate 3-phosphatase and dual-specificity protein phosphatase; PTK2, protein tyrosine kinase 2; SLC39A7, zinc transporter SLC39A7.

1,225.60, 1,363.85, 1,688.71, 1789.52, 2020.21, and 2156.42 Da were found to be part of the proteins CLCN1, CLUL1, EDNRA, PTEN, SLC39A7, and CLN6, respectively. CLCN1 is the voltage-gated chloride ion channel protein. It is located in the cell membrane and is known to be involved in the repolarization of skeletal muscle cells [48]. CLUL1 is a clusterin-like protein. Its alternative name is retinal-specific clusterin-like protein because it is mainly localized in retinal cone photoreceptor cells [49]. However, its molecular function in dogs is still unclear. Furthermore, EDNRA is an endothelin-1 receptor that can be classified into 2 subtypes: endothelin receptor type A ($ET_A$) [50], and endothelin receptor type B ($ET_B$) [51]. $ET_A$ is predominantly located on vascular smooth muscle cells. $ET_B$ is mostly abundant on endothelial

cells, followed by smooth muscle cells to some extent. When endothelin-1 (ET-1) activates both receptors on vascular smooth muscle cells, vasoconstriction is induced, and smooth muscle cells proliferate. Activation of $ET_B$ on endothelial cells causes vasodilation, and the clearance of ET-1 [52]. In addition, PTEN has been reported to be a tumour suppressor gene. It translates dual-specificity protein phosphatase that is involved in tumour suppression. Moreover, PTEN mutation is frequently found to be associated with many human cancers, such as endometrial carcinomas, breast carcinomas, prostate carcinomas and gliomas [53], as well as canine osteosarcoma cell lines and tumours [54]. SLC39A7 is a zinc transporter that transports zinc across the cytoplasm into the cell from the extracellular compartment or transports zinc into the cell from intracellular organelles, such as the endoplasmic reticulum (ER), Golgi apparatus and mitochondria [55]. CLN6 is a ceroid-lipofuscinosis neuronal protein 6 homologue. A mutation of CLN6 would result in neuronal ceroid lipofuscinoses (NCLs), the most common neurodegenerative disease, which primarily occur in children [56]. The detection of CLN6-induced NCL has also been reported in some breeds of dogs, including Border Collies, English Setters, American Bulldogs, Dachshunds, and Australian Shepherds [57]. Four of the 6 proteins, including CLCN1, EDNRA, PTEN and SLC39A7, were associated with cardiovascular drugs, as demonstrated in the protein network interaction analysis (Fig 4). If the peptide candidates obtained from this study were derived from the degradation of these specified proteins, it was suggested that they might involve in the pathogenesis of PH and MVD as predicted by the Stitch program. SLC39A7 and CLUL1 showed no relationship with cardiovascular drugs or other proteins in the pathway. This possibly suggests that these 2 proteins could be novel candidate biomarkers whose association with cardiovascular disorders has never before been identified.

Differential protein expression has been reported in the CKCS dogs affected by MVD with different severity levels by MALDI-TOF MS using serum samples [58]. The candidate peptides obtained from this current study were all different from those in the previous study, probably due to the different peptide identification techniques (MALDI-TOF/TOF vs. LC-MS/MS) and different breeds of dogs enrolled (CKCS vs. no limitation on breed).

This study demonstrates a promising finding for the diagnosis of MVD and PH in dogs using MALDI-TOF MS combined with LC-MS/MS. However, there are several limitations to this study. First, there was a small sample size in the MVD B PH group. This group consists of dogs with MVD stage B2 and secondary PH. Since MVD stage B2 is an asymptomatic stage, the left atrial pressure that would affect the pulmonary circulation is expected to be lower than that in the symptomatic stage of MVD (MVD stage C2). Hence, secondary PH is less likely to develop in dogs with MVD stage B2 than those with MVD stage C2. For this reason, fewer samples were collected for the MVD B PH group than for the other groups. Second, several factors, including age, sex, breed and previous treatment, could not be controlled. These factors could confound the outcome of the study. Nevertheless, our results obtained using pooled samples show different peptide barcode patterns among all 5 groups. This indicates that these factors would not affect the results of this study. Additionally, the results from individual samples in each group were obtained using 3D-PCA before the pooled samples were analyzed to exclude samples with peptide expression that deviated from most of the samples within the group. Third, the median age of the normal group was still a half younger than the other groups. The lack of aging dogs with no concurrent systemic disorders was the cause of this situation. As a result, the majority of dogs in the normal group were substantially younger than those in the MVD and PH groups. For this reason, we cannot completely assume that peptide barcodes could entirely distinguish healthy dogs from those with heart diseases. Because if the normal group gets older, there is still a chance that some peptide barcodes from the normal group could interact with the group with heart diseases. However, among the heart disease

group, the result demonstrates that peptide barcodes could promisingly differentiate dogs affected by asymptomatic and symptomatic stages of MVD with and without PH. Finally, MALDI-TOF and LC-MS are now difficult to access in veterinary practice. Because a clinic or hospital would definitely be unable to afford such equipment, thus in routine practice, sample analysis for peptide barcodes would currently have to be outsourced. In contrast to human medicine, MALDI-TOF MS is now widely used as an early diagnostic and monitoring tool, particularly in the field of neoplastic diseases [19–27]. As a result, if we actually want to use the MALDI-TOF method to diagnose dogs with MVD and PH, we might as well submit the samples to a laboratory center that responsible for human proteomic analysis. Despite its current practical limitations, this study could add a point of view in the proteomic field of veterinary cardiology. In the future, the findings of this study could contribute to the development of a new diagnostic approach for dogs with MVD and PH.

This study was considered a preliminary study, and further investigations with more samples from dogs affected by MVD stage B2 with PH, as well as more dogs with the same heart disease condition but with different ages, sexes and breeds, are expected to be performed to support and verify the results of the current study. However, the different peptide barcodes among all 5 groups initially confirm the differences in peptide expression among dogs with different heart disease conditions. Further studies on peptide and protein expression in dogs with MVD and PH are recommended.

## Conclusion

This present study shows an advantage of peptide barcodes using MALDI-TOF MS in providing useful information for the diagnosis of MVD affected by different severities with and without PH. In combination with LC-MS/MS, MALDI-TOF MS revealed candidate peptides associated with the disease conditions. In this study, different peptide barcodes and discrete clusters of peptides were expressed among all groups, as determined by the 3D-PCA method. This indicates that MALDI-TOF MS has promise as an optional or additional technique for the detection of canine MVD and PH with different clinical conditions. The obtained outcome of specific peptide barcodes in dogs with different types of heart diseases may contribute to the development of novel diagnostic methods for patients with heart diseases. These findings appear to be helpful for the future diagnosis of many diseases in veterinary medicine. However, more information from the numbers of samples in dogs with other diseases, apart from heart disease, is required to identify peptide barcodes to establish a large database of peptide barcode libraries in veterinary medicine. Then, this method could be used as a rapid test for the diagnosis and discrimination of asymptomatic and symptomatic MVD with and without PH, as well as other important canine diseases in the future.

## Acknowledgments

We sincerely thank all staff at Functional Proteomics Technology Laboratory, National Center for Genetic Engineering and Biotechnology (BIOTEC), Pathum Thani, Thailand, for laboratory assistance and support. We would like to thank all staff at the laboratory unit, Prasu-Arthorn Animal Hospital, Faculty of Veterinary Science, Mahidol University, Nakhon Pathom, Thailand, for assistance in specimen processing and storage. Special thanks to Dr. Nathamon Yimpring and Dr. Sekkarin Ploypetch for technical advice during the study.

## Author Contributions

**Conceptualization:** Sittiruk Roytrakul, Walasinee Sakcamduang.

**Data curation:** Nattapon Riengvirodkij.

**Formal analysis:** Walasinee Sakcamduang.

**Funding acquisition:** Sittiruk Roytrakul, Walasinee Sakcamduang.

**Investigation:** Nattapon Riengvirodkij, Sittiruk Roytrakul, Janthima Jaresitthikunchai, Narumon Phaonakrop, Sawanya Charoenlappanich, Walasinee Sakcamduang.

**Methodology:** Nattapon Riengvirodkij, Sittiruk Roytrakul, Janthima Jaresitthikunchai, Narumon Phaonakrop, Sawanya Charoenlappanich, Walasinee Sakcamduang.

**Project administration:** Sittiruk Roytrakul, Walasinee Sakcamduang.

**Resources:** Nattapon Riengvirodkij, Sittiruk Roytrakul, Janthima Jaresitthikunchai, Narumon Phaonakrop, Sawanya Charoenlappanich, Walasinee Sakcamduang.

**Software:** Sittiruk Roytrakul, Janthima Jaresitthikunchai, Narumon Phaonakrop, Sawanya Charoenlappanich.

**Supervision:** Sittiruk Roytrakul, Walasinee Sakcamduang.

**Validation:** Nattapon Riengvirodkij, Sittiruk Roytrakul, Walasinee Sakcamduang.

**Visualization:** Nattapon Riengvirodkij, Sittiruk Roytrakul, Walasinee Sakcamduang.

**Writing – original draft:** Nattapon Riengvirodkij.

**Writing – review & editing:** Nattapon Riengvirodkij, Sittiruk Roytrakul, Walasinee Sakcamduang.

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
