## [Decision Letter · Decision Letter 0]

24 May 2021

PONE-D-21-09311

Peptide barcodes in dogs affected by mitral valve disease with and without pulmonary hypertension using MALDI - TOF MS and LC - MS / MS

PLOS ONE

Dear Dr. Sakcamduang,

Thank you for submitting your manuscript to PLOS ONE. After careful consideration, we feel that it has merit but does not fully meet PLOS ONE’s publication criteria as it currently stands. Therefore, we invite you to submit a revised version of the manuscript that addresses the points raised during the review process.

We look forward to receiving your revised manuscript.

Kind regards,

Joseph Banoub, Ph,D., D. Sc., FCIC, FRSC.

Academic Editor

PLOS ONE

Journal Requirements:

3. Please upload a copy of Supporting Information File which you refer to in your text (line 442).

Reviewers' comments:

Reviewer's Responses to Questions

**Comments to the Author**

1. Is the manuscript technically sound, and do the data support the conclusions?

Reviewer #1: Yes

Reviewer #2: Yes

2. Has the statistical analysis been performed appropriately and rigorously? 

Reviewer #1: I Don't Know

Reviewer #2: Yes

3. Have the authors made all data underlying the findings in their manuscript fully available?

Reviewer #1: Yes

Reviewer #2: Yes

4. Is the manuscript presented in an intelligible fashion and written in standard English?

Reviewer #1: Yes

Reviewer #2: Yes

5. Review Comments to the Author

Reviewer #1: The authors in this manuscript used mass spectrometry-based proteomics to differentiate between the different stages of mitral valve disease (MVD) found in dogs. Matrix-assisted laser desorption/ionization time-of-flight mass spectrometry (MALDI-TOF MS) and liquid chromatography-tandem mass spectrometry (LC-MS/MS) was used to compare the serum proteomes from healthy dogs, dogs with MVD stage B (MVD B, asymptomatic stage), MVD stage C (MVD C, symptomatic stage), MVD stage B with PH (MVD B PH) and MVD stage C with PH (MVD C PH). This resulted in the identification of the differences in the serum peptide barcodes of dogs in the MVD B, MVD C, MVD B PH, MVD C PH, and normal control groups. Six discriminatory peptides at m/z 1,225.60, 1,363.85, 1,688.71, 1789.52, 2020.21, and 2156.42 were altered in their abundance, representing changes in the protein levels of peptide barcode components among the groups. These results are helpful in the diagnosis of different stages of MVD with PH or without PH (pulmonary hypertension). Overall, I recommend the publication of this work as it demonstrates the application of mass spectrometry-based proteomics in diagnosis.

Reviewer #2: The manuscript by Riengvirodkij et al., titled “Peptide barcodes in dogs affected by mitral valve disease with and without pulmonary hypertension using MALDI-TOF MS and LC-MS/MS”, uses mass spectrometry to examine peptides specific to mitral valve disease, without or with pulmonary hypertension. It is an interesting approach, and it appears to be quite accurate in distinguishing MVD without or with hypertension. Specific comments are as follows:

1. Of concern is the choice of the ‘normal’ group, having a median age of about half that of the other groups. Is it possible that some of the barcodes identified could become non-specific to one or more groups if the ‘normal’ group had an increased age? A discussion point on the choice of the ‘normal’ group and limitations of this group with age is needed.

2. A practical limitation would include the access to MALDI and LC equipment to a veterinary clinic. It is very unlikely a clinic would purchase such equipment, thus in several instances the analyses of samples would have to be outsourced. Consider a discussion point about this limitation.

3. Was pro B-type-natriuretic peptide data available for the animals examined? If so, was it also examined via mass spectrometry, and how did the differing methods compare?

4. Minor: though well written, some correction to punctuation is needed through the manuscript.

6. PLOS authors have the option to publish the peer review history of their article (what does this mean?). If published, this will include your full peer review and any attached files.

Reviewer #1: No

Reviewer #2: No

---

## [Author Response · Author response to Decision Letter 0]

2 Jul 2021

Journal Requirement:

We have checked our references and found none of them have been retracted. We deleted reference 11 as it duplicated reference 6 in the previous manuscript (line 519-520 of the track-change file). The revision manuscript then has 58 references relevant the PLOS ONE’s requirements.

Reviewer 1: 

There is no specific concern from reviewer 1. Thank you very much for your consideration recommending the publication of our study.

Reviewer 2:

1. Of concern is the choice of the ‘normal’ group, having a median age of about half that of the other groups. Is it possible that some of the barcodes identified could become non-specific to one or more groups if the ‘normal’ group had an increased age? A discussion point on the choice of the ‘normal’ group and limitations of this group with age is needed.

Because mitral valve disease and pulmonary hypertension, the conditions we have focused on in this study, are commonly found in older dogs, usually more than 5 years old. Accordingly, we designed to recruit dogs at least 5 years old to match the age between normal and the disease groups. However, despite this attempt, the median age of the normal group was still a half younger than the other groups. The reason for this situation was the lack of aging dogs without any concurrent systemic diseases. Therefore, most of the enrolled dogs in the normal group were much younger than dogs in the group with mitral valve disease and pulmonary hypertension. For this concern, it is still possible that some peptide barcodes of the normal group may become non-specific to the disease group if the normal group has an increased age. This limitation has been additionally discussed as the study’s limitations in the discussion part (line 407-420 of the track-change file, and line 406-420 of the non-track-change file).

2. A practical limitation would include the access to MALDI and LC equipment to a veterinary clinic. It is very unlikely a clinic would purchase such equipment, thus in several instances the analyses of samples would have to be outsourced. Consider a discussion point about this limitation.

Presently, MALDI-TOF MS is being used in human medicine as a common diagnostic test for early diagnosis and monitoring, particularly in the field of neoplastic illnesses. It has been used to detect cancer in its early stages in a variety of cancers. Conversely, in veterinary medicine, there is a limitation in reaching MALDI-TOF and LC mass spectrometry in routine practice due to the high cost of the equipment. However, this study could add a point of view in the proteomic field of veterinary cardiology. Although the use of MALDI-TOF as a diagnostic method in dogs with heart diseases is still currently limited in the present day. In the future, the findings of this study may assist in the development of a novel diagnostic method for dogs with heart diseases. We have already added the discussion of this point as a study limitation (line 420-429 of the track-change file, 420-428 of the non-track-change file).

3. Was pro B-type-natriuretic peptide data available for the animals examined? If so, was it also examined via mass spectrometry, and how did the differing methods compare?

We have no data available of N-terminal B-type-natriuretic peptides for the enrolled animals in this study. To our knowledge, no study reported on comparing the measurement of N-terminal B-type-natriuretic peptide levels and mass spectrometry as diagnostic methods in dogs with mitral valve disease and pulmonary hypertension. This is an interesting topic that recommends further investigation.

4. Minor: though well written, some correction to punctuation is needed through the manuscript.

We thoroughly revised the language usage and punctuation in our manuscript.

---

## [Decision Letter · Decision Letter 1]

21 Jul 2021

Peptide barcodes in dogs affected by mitral valve disease with and without pulmonary hypertension using MALDI-TOF MS and LC-MS/MS

PONE-D-21-09311R1

Dear Dr. Sakcamduang,

We’re pleased to inform you that your manuscript has been judged scientifically suitable for publication and will be formally accepted for publication once it meets all outstanding technical requirements.

Kind regards,

Joseph Banoub, Ph,D., D. Sc.

Academic Editor

PLOS ONE

Additional Editor Comments (optional):

Reviewers' comments:

Reviewer's Responses to Questions

**Comments to the Author**

1. If the authors have adequately addressed your comments raised in a previous round of review and you feel that this manuscript is now acceptable for publication, you may indicate that here to bypass the “Comments to the Author” section, enter your conflict of interest statement in the “Confidential to Editor” section, and submit your "Accept" recommendation.

Reviewer #2: All comments have been addressed

2. Is the manuscript technically sound, and do the data support the conclusions?

Reviewer #2: Yes

3. Has the statistical analysis been performed appropriately and rigorously? 

Reviewer #2: Yes

4. Have the authors made all data underlying the findings in their manuscript fully available?

Reviewer #2: Yes

5. Is the manuscript presented in an intelligible fashion and written in standard English?

Reviewer #2: Yes

6. Review Comments to the Author

Reviewer #2: All comments were addressed; apologies to the authors, as this reviewer missed a commented point in the Discussion.

7. PLOS authors have the option to publish the peer review history of their article (what does this mean?). If published, this will include your full peer review and any attached files.

Reviewer #2: No

---

## [Editor Report · Acceptance letter]

29 Jul 2021

PONE-D-21-09311R1 

Peptide barcodes in dogs affected by mitral valve disease with and without pulmonary hypertension using MALDI-TOF MS and LC-MS/MS 

Dear Dr. Sakcamduang:

I'm pleased to inform you that your manuscript has been deemed suitable for publication in PLOS ONE. Congratulations! Your manuscript is now with our production department. 

Kind regards, 

on behalf of

Dr. Joseph Banoub 

Academic Editor

PLOS ONE